# Dissipation and Residue Pattern of Dinotefuran, Fluazinam, Indoxacarb, and Thiacloprid in Fresh and Processed Persimmon Using LC-MS/MS

**DOI:** 10.3390/foods11030416

**Published:** 2022-01-31

**Authors:** Hyun-Ho Noh, Seung-Hyeon Jo, Hyeon-Woo Shin, Dong-Ju Kim, Young-Jin Ham, Jun-Young Kim, Dan-Bi Kim, Hye-Young Kwon, Kee-Sung Kyung

**Affiliations:** 1Residual Agrochemical Assessment Division, Department of Agro-Food Safety and Crop Protection, National Institute of Agricultural Sciences, Wanju 55365, Korea; noh1983@korea.kr (H.-H.N.); danbi6334@korea.kr (D.-B.K.); 2Jeonbuk Analytical Research Group, Translational Toxicology Research Division, Korea Institute of Toxicology, Jeongeup 56212, Korea; seunghyeon.jo@kitox.re.kr; 3Environmental Toxicology & Chemistry Center, Korea Testing & Research Institute, Hwasun 58141, Korea; hyeonwoo@ktr.or.kr; 4Department of Environmental and Biological Chemistry, College of Agriculture, Life and Environment Science, Chungbuk National University, Cheongju 28644, Korea; kimdj6746@naver.com (D.-J.K.); youngjin0223@naver.com (Y.-J.H.); tim8518@naver.com (J.-Y.K.); 5Department of Digital Agriculture, Rural Development Administration, Jeonju 54875, Korea; kwonhy91@korea.kr

**Keywords:** pesticide residue, persimmon, processing factor, reduction factor, LC-MS/MS

## Abstract

Pesticides which are diluted and sprayed according to the pre-harvest interval (PHI) are generally decomposed and lost through various factors and pathways, and the leftover pesticides are known as residual pesticides. This study aims to determine the dissipation of residual amounts of dinotefuran, fluazinam, indoxacarb, and thiacloprid in persimmon and the changes in the concentration of various processing products. Pesticide spraying is performed in accordance with the GAP (good agricultue practice) of Korea, and the processed products are manufactured using a conventional method after removing the skin of persimmons. The modified QuEchERS (Quick, Easy, Cheap, Effective, Rugged, and Safe) method and an optimized method using LC-MS/MS (liquid chromatography mass spectrometry) is implemented to analyze the residual pesticides. The linearity, recovery, and LOQ (limit of quantitation) are presented to verify the analysis method. The amount of residual pesticides tested decreases significantly in a time-dependent manner, regardless of the minimal dilution effect present due to growth. The residual concentration does not vary significantly during the processing stage despite the removal of the systemic pesticides, dinotefuran and thiacloprid. The residues of non-systemic pesticides, fluazinam and indoxacarb, are typically removed by the peeling removal and processing methods. The reduction factor of dinotefuran, whose residual concentration is increased, is less than 1, and the absolute amount of pesticides is decreased through processing. The results of this study can be used as the scientific basis data to ensure the safety of residual pesticides in processed products in the future.

## 1. Introduction

Pesticides are essential agricultural materials that can protect crops from various pests and weeds; this ensures that high-quality agricultural products can be provided to consumers, ensuring stable income for the farmers [1]. However, pesticides are toxic organic compounds, whose toxicity varies based on the amount [2]; their usage continues to increase according to recent reports [3]. Exposure to pesticides can result in various harmful side effects in humans [4]. Therefore, the pre-harvest interval (PHI) and maximum residue limit (MRL) are set for pesticides according to the situation of each country to manage the pesticide usage and to reduce the risks involved, thus improving the safety [5]. EU-harmonized MRLs are set for more than 1300 pesticides covering 378 food products/food groups. A default MRL of 0.01 mg kg^−1^ is applicable to nearly 690 of these pesticides, which are not explicitly mentioned in the MRL legislation [6].

Pesticides which are diluted and sprayed according to the PHI are generally decomposed and lost through various factors and pathways, and the leftover pesticides are known as residual pesticides. The concentration of residual pesticides is determined by the spraying factors (concentration of spraying solution, frequency, etc.), environmental factors (rainfall, sunlight, temperature, etc.), and morphological factors (shape, canopy, etc.) [5,7].

From a consumer’s point of view, washing agricultural products in running water or stagnant water and cooking them is the first step in food processing. The residual pesticides can be removed in the processing stage [8]. Additionally, most residual pesticides that have not been translocated into the agri-food can be removed during the peeling process. The residual pesticides can be reduced during boiling or juicing, and the amount of residual pesticides may decrease based on the drying method [9,10,11]. However, the reduction patterns may vary depending on the chemical properties of the pesticides.

Persimmon is a deciduous fruit tree believed to have originated from China and is grown primarily in the East Asian countries such as Korea, China, and Japan. However, it has also been introduced and cultivated in temperate climate regions such as California, Italy, Israel, Australia, and New Zealand. Persimmon is an excellent source of carotene, potassium, and vitamin C [12]. Persimmons can either be consumed fresh or through various drying process. There are three typical drying processes. The first is “dried persimmon”, in which the fruit is first washed and then completely dried after removing the peels. This is the most representative and traditional persimmon processed product. The second is the “semi dried persimmon”, which is only half-dried after washing and peeling. This processed product is popular with consumers for its high sugar content and soft texture. The last process is the “dried persimmon slice”, in which the washed and peeled persimmon is dried after it is sliced [12]. It is a processed product that has strengths in portability and convenience. It is an excellent material to determine the changes in the residual pesticides by using the various drying methods.

Persimmon is primarily cultivated in certain regions in Korea, and open-field fruits shipped in late autumn are generally popular. Some of them are exported to Southeast Asia and the United States. Additionally, there are cases where MRLs are set in Korea while they are not set in the United States since the pattern of pesticide usage in the two countries varies. Among them, the MRLs for dinotefuran, fluazinam, indoxacarb, and thiacloprid in persimmon in Korea were set by the Ministry of Food and Drug Safety (MFDS). However, they have not been established in the United States, which means that the United States has a zero tolerance towards the use of pesticides [13]. Therefore, the use of such pesticides in Korea may lead to the refusal of customs clearance due to the presence of residual pesticides during the United States customs clearance process, which presents a major obstacle to trade between Korea and the United States. These problems are not limited to this case and can also be applied to different crops from other countries [14].

Additionally, the demand for processed foods has been increasing in recent years [15], and it is crucial to thoroughly analyze the movements of residual pesticides to ensure the safety of agricultural products. Particularly, the research conducted on persimmon processing, which is mostly manufactured as processed products, has been insufficient. Therefore, this study aims to analyze the changes in residual pesticides while manufacturing persimmons into various processed products and to ensure the safety of consumers by providing clear evidence of the absence of residual pesticides in persimmon products consumed in the United States, Europe, and Asia.

## 2. Materials and Methods

### 2.1. Field Trials

The tested pesticides, dinotefuran, fluazinam, indoxacarb, and thiacloprid, are widely used in Korea to control pests in various crops and are used according to a set MRL value in Korea. The pesticides diluted 2000 times were sprayed onto the persimmon crop under immature conditions 14 days before harvest with a backpack sprayer. The solution composition is as follows: dinotefuran 20% water dispersible granule (WG), fluazinam 40% wettable powder (WP), indoxacarb 10% WP, and thiacloprid 10% suspension concentrate (SC). Table 1 presents the detailed information on pesticide spraying.

The field trial was conducted at three locations: Cheongdo-gun (35°36′40.7″ N 128°36′51.2″ E, site 1), Sangju-si (36°24′10.2″ N 128°14′25.2″ E, site 2), and Jinju-si (35°10′37.3″ N 128°09′32.3″ E, site 3), which produce approximately 66% of the persimmon in Korea [16]. Persimmons are divided into sweet and astringent persimmons, and the astringent persimmons are used to manufacture the processed products. The astringent persimmons for processing materials were collected from sites 1 and 2. Additionally, fresh samples of sweet persimmons were collected from site 3 and analyzed, which were not processed. The pattern of residual pesticides in fresh persimmons were analyzed in a time-dependent manner at all the sites after spraying pesticide according to the PHI on the plot of scale, 25.4 m W. × 10.1 m L. The samples were collected after 3 h on days 0, 1, 3, 5, 7, and 14 (decline study). The samples on day 14 were processed to dried persimmon slices, semi-dried persimmon, and dried persimmon (harvest study) on the pre-harvest day of PHI. The processed products were manufactured at Nature Farm Agricultural Co., Ltd. (Cheondo, Gyeongbuk, Korea), and Figure 1 presents the details of the processing stage. The prepared samples, fresh persimmon, and processing products were blended with dry ice and then stored in a freezer at −20 °C.

### 2.2. Reagents and Materials

Analytical grade dinotefuran (98.5% purity), DN (98.5% purity), MNG (96.22% purity), UF (99.8% purity), fluazinam (99.89% purity), indoxacarb (96.1% purity), and thiacloprid (99.9% purity) were obtained from Dr. Ehrenstofer GmbH (Augsburg, Germany). Acetonitrile (HPLC grade) for standard dilutions, sample extraction, and the LC mobile phase were obtained from Merck (Darmstadt, Germany). HPLC-grade formic acid (> 98%) for the LC mobile phase was purchased from Merck (Darmstadt, Germany). Lead(II) acetate trihydrate was purchased from Daejung Chemicals (Iksan, Korea). The QuEChERS extraction packets and d-SPE tubes were obtained from Agilent Technologies (Santa Clara, CA, USA). Combi-514R (Hanil Scientific Inc., Gimpo, Korea) and Combi-408 (Hanil Scientific Inc., Gimpo Korea) centrifuges were used. A 2010 Geno/Grinder^®^ automated homogenizer (SPEX SamplePrep LLC., Metuchen, NJ, USA) was used for sample extraction. The rotary evaporator used EYELA N-1000 (EYELA, Tokyo, Japan).

### 2.3. Sample Preparation

#### 2.3.1. Dinotefuran and Its Metabolites (DN, MNG and UF)

Fresh persimmon of 5 g and its processed products were weighed in a tall beaker (300 mL), and 100 mL of methanol-distilled water solution (9:1, *v*/*v*) was added. The mixture was homogenized at 10,000 rpm for 5 min in a blender, and the homogenate was filtered under the vacuum condition. The flask and filter cake were rinsed with 50 mL of fresh methanol, and the rinsate was combined with the previous filtrate. The organic solvent extract was evaporated to dryness at 35 °C using a rotary vacuum evaporator. The residue was redissolved in 10 mL of water, and the redissolved solution was filtered and used for LC-MS/MS analysis.

#### 2.3.2. Fluazinam, Indoxacarb, and Thiacloprid

The pesticide residue was analyzed using the modified QuEChERS method [11]. The pesticide residues in the samples were extracted using acetonitrile and separated using the QuEChERS EN extraction method (4 g of MgSO_4_, 1 g of NaCl, 1 g of trisodium citrate dihydrate and 0.5 g of disodium citrate sesquihydrate). The processing products were wetted using distilled water to increase the extraction efficiency, and a saturated lead acetate solution was added to separate the sample and the solvent, along with the salts. The solvent with the extract was purified by using the d-SPE (dispersive-solid phase extraction) method containing MgSO_4_, primary secondary amine (PSA), and graphite carbon black. Figure 2 presents the detailed information of the residual pesticide analysis method.

### 2.4. Instrumental Analysis

The residual pesticide was analyzed using LC-MS/MS (Nexera X2 equipped with LCMS-8050, Shimadzu Corporation, Kyoto, Japan). The column and mobile phase compositions were applied based on the target pesticides. The column for the analysis of dinotefuran including its metabolites in the samples was Phenomenex Kinetex^®^ C18 (2.1 mm i.d. × 150 mm L., 2.6 μm particle size). Acquity UHPLC BEH C18 (2.1 mm i.d. × 100 mm L., 1.7 μm particle size) was used for fluazinam residue analysis. Indoxacarb and thiacloprid were used with Shim-Pack GIST-HP C18 (2.1 mm i.d. × 150 mm L., 3.0 μm particle size).

Dinotefuran and its metabolites were analyzed using acetonitrile and distilled water under gradient conditions to ensure the efficiency of compound separation. Initially, 100% of the distilled water was allowed to flow for 2 min and was gradually changed to 50% for 6 min; it was then held for 7 min. Subsequently, it was adjusted to the initial conditions and fixed at 100% for 10 min for stabilization. Fluazinam, indoxacarb, and thiacloprid were analyzed using an isocratic elution method. Fluazinam was analyzed as a mixed solvent with a ratio of 10:90 (*v*/*v*) of distilled water containing 0.1% formic acid and acetonitrile containing 0.1% formic acid. The mixture solvent of the mobile phase for indoxacarb included distilled water containing 0.1% formic acid and acetonitrile containing 0.1% formic acid in the ratio of 30:70 (*v*/*v*). Thiacloprid used a solvent in which distilled water and methanol were mixed at a ratio of 5:95 (*v*/*v*). The flow rates of the mobile phase were 0.2 mL min^−1^ for fluazinam and 0.3 mL min^−1^ for the others. The electrospray ionization (ESI) mode was adopted in the tandem MS condition, and fluazinam was analyzed as the negative mode while the rest were analyzed in the positive mode. The flow rates of the drying gas, heating gas, and nebulizing gas were 10 L min^−1^, 10 L min^−1^, and 3 L min^−1^, respectively. The source temperature was 150 °C, and the spray voltage was 3.0 kV. Multiple reaction monitoring (MRM) was applied to all the target pesticides, and LabSolutions (version 5.93, Shimadzu Coperation, Kyoto, Japan) was used to process the quantitative results of the MRM data. Table 2 lists the detailed MRM conditions of the target pesticide for instrumental analysis.

### 2.5. Method Validation

The LOQs for the test pesticides were determined as the minimum concentration providing a signal to noise (S/N) ratio of 10 or more on the chromatogram, along with a reasonable recovery precision (relative standard deviation, RSD 20%). The calibration curves for the quantitation of the pesticide residues in the test samples were obtained from matrix-matched standards, and the linearity of calibration was expressed as a correlation coefficient (r^2^) at a weighting factor of 1/*x*. The recovery was evaluated at three fortification levels: LOQ, 10 LOQ, and Korean MRL. For each fortification level, the accuracy was expressed as the average of the recovery rates (*n* = 5) and the precision was expressed by its RSD. The effectiveness of the analysis method was determined by comparing the recovery test results with the SANTE/12682/2019 [17]. The FAO guidelines present an efficient range of recovery and RSD based on the fortification levels. In this case, if the fortification level is ≤0.001 mg kg^−1^, and the resultant recovery range and RSD are 50–120% and <35%, respectively; if >0.001 or ≤0.01 mg kg^−1^, then 60–120% and <30%, respectively; if 0.01 of 0.1 mg kg^−1^, then 70–120% and <20%, respectively; and if >0.1 or ≤1.0 and >1.0 mg kg^−1^, then 70–1≤10% and <15% and <10%, respectively [18].

### 2.6. Residue Definition of Test Pesticides

Codex only requires the results of the parent compound if it is an analysis to establish the MRL of dinotefuran in plants. For the estimation of the dietary intake, the sum of dinotefuran, DN, and UF must be expressed as dinotefuran in Codex. However, the results for another metabolite, MNG are also presented. The residual amount of dinotefuran was calculated by multiplying the molecular weight ratio of the parent compound and each metabolite by the residual amount of the metabolite and then summing it up using Equation (1) [19] as follows:R_TD_ = R_D_ + (R_DN_ × CF_DN_) + (R_UF_ × CF_UF_) + (R_MNG_ × CF_MNG_),(1)
where R_TD_, R_D_, R_DN_, R_UF_, and R_MNG_ are the total dinotefuran, dinotefuran, DN, UF, and MNG residues (mg kg^−1^), respectively, and CF_DN_, CF_UF_, and CF_MNG_ are the conversion factors to calculate the ratio of the molecular weight of dinotefuran and DN, UF, and MNG, respectively.

Indoxacarb has R- and S-form structures, but it is used to analyze only the R-form according to Codex. It is necessary to analyze the parent compound in the case of thiacloprid [20]. Fluazinam did not provide a residual definition in Codex, and it has been suggested that only the parent compound is analyzed by EFSA [21].

### 2.7. Half-Lives of Test Pesticide in Fresh Persimmon

The dissipation patterns of the test pesticides in fresh persimmon over time were evaluated using a first-order kinetic model, and the dissipation dynamic equation and half-life were calculated using Equations (2) and (3), respectively, as follows:C_t_ = C_0_e^−kt^,(2)
t_1/2_ = ln(2)/k,(3)
where C_t_ (mg kg^−1^) represents the concentration of the residual pesticides at time t (day), C_0_ (mg kg^−1^) represents the initial concentration of the residual pesticide at time t = 0, and k (day^−1^) represents the degradation coefficient [11,22].

### 2.8. Processing and Reduction Factors

The processing factor (PF) is an index which indicates that if the residual concentration ratio before and after processing is 1 or more, the residual concentration of the pesticides increases during the processing stage [18]. This can also be used as a basis to establish MRLs among the processed products. The reduction factor (RF) is an index that can determine the amount of reduction in the absolute amount of pesticides during the processing stage [11]. For example, if the reduction coefficient was 0.5, the absolute amount of pesticides is decreased by 50% during the processing stage. The manufacturing yield of the processed product must be determined to perform this calculation, and the moisture content of the raw material and the processed product must be measured. Additionally, the reduction factor can be calculated by considering the residual amount and yield before and after processing, based on the dry weight [11,18], as shown below:(4)$$\mathrm{PF}=\frac{\mathrm{Residue}\;\mathrm{in}\;\mathrm{raw}\;\mathrm{product}\;\left(\mathrm{mg}/\mathrm{kg}\right)}{\mathrm{Residue}\;\mathrm{in}\;\mathrm{processing}\;\mathrm{product}\;\left(\mathrm{mg}/\mathrm{kg}\right)},$$
(5)$$\mathrm{Residue}\;\mathrm{in}\;\mathrm{raw}\;\mathrm{product}\;\mathrm{based}\;\mathrm{on}\;\mathrm{dry}\;\mathrm{weight}\;\left(\mathrm{mg}/\mathrm{kg}\right)=\;\frac{100\times \mathrm{residue}\;\mathrm{in}\;\mathrm{raw}\;\mathrm{product}\;\left(\mathrm{mg}/\mathrm{kg}\right)}{100-\mathrm{water}\;\mathrm{content}\;\mathrm{of}\;\mathrm{raw}\;\mathrm{product}\;\left(\%\right)},$$
(6)$$\mathrm{RF}=\frac{\mathrm{Residue}\;\mathrm{in}\;\mathrm{processing}\;\mathrm{product}\;\mathrm{based}\;\mathrm{on}\;\mathrm{dry}\;\mathrm{weight}\;\left(\mathrm{mg}/\mathrm{kg}\right)}{\mathrm{Residue}\;\mathrm{in}\;\mathrm{raw}\;\mathrm{product}\;\mathrm{based}\;\mathrm{on}\;\mathrm{dry}\;\mathrm{weight}\;\left(\mathrm{mg}/\mathrm{kg}\right)},$$

### 2.9. Statistical Analysis

One-way analysis of variance (ANOVA) and *t*-test of the pesticide residues were performed using SPSS v. 26 (IBM Corp., Armonk, NY, USA). Duncan’s multiple range test was performed at *p* < 0.05 to identify the significant differences between the treatment methods.

## 3. Results and Discussion

### 3.1. Validation of the Analytical Method

The LOQ of the test pesticides in the analyte was 0.01 mg kg^−1^, which has an S/N ratio of 10 or more. The linearity of the calibration curve among the samples expressed in the correlation coefficient (r^2^) used to quantify the test pesticides was 0.99 or higher, all of which satisfied SANTE/12682/2019 [17], and chromatograms of the matrix-matched standard were presented in Figure 3. Table 3 presents the summarized results. A recovery test was conducted to validate the analysis method for the pesticide residue, and the results are presented in Figure 4. The average recovery and RSD of the test pesticide, dinotefuran, including its metabolite DN, MNG, and UF, fluazinam, indoxacarb, and thiacloprid, in fresh persimmon for all the fortification concentrations were found to be 73.0–106% and 0.510–8.85%, respectively. In dried persimmon, it was calculated to be from 71.7% to 117% for recovery and from 0.940% to 9.76% for RSD. The average recovery and RSD in dried persimmon slice were found to be 71.1–117% and 1.17–9.18%, respectively. The average recovery and RSD for the semi-dried persimmon ranged from 71.0% to 103% and 1.12% and 13.0%, respectively. Consequently, the results of the recovery test conducted in this study, which conform to this criterion by SANTE/12682/2019 were obtained in all cases.

### 3.2. Results of Pesticide Residue Analysis

#### 3.2.1. Dissipation Pattern of Test Pesticide in Fresh Persimmon

The residual amount of dinotefuran, the parent compound in Site 1, was 0.47 ± 0.04 mg kg^−1^ on the day of the final pesticide spraying and 0.28 ± 0.01 mg kg^−1^ on Day 14. The residual amounts for the initial concentration (0 day) of Sites 2 and 3 were 0.53 ± 0.03 and 0.46 ± 0.01 mg kg^−1^, respectively, and for day 14, were 0.24 ± 0.02 and 0.27 ± 0.03 mg kg^−1^, respectively. In the case of metabolite DN, all the samples presented residual amounts below the LOQ. MNG was not detected in the initially collected samples but was detected at Site 1 on day 14 and at Site 2 from day 7. MNG was not detected at Site 3 in this study. UF was detected at the LOQ level from the seventh day of the final pesticide spraying at all the sites. Table 4 presents the total residual amount of dinotefuran, which can be summed by converting the residual amount of metabolites into a parent compound.

The total residual amounts of dinotefuran on day 0 were 0.47 ± 0.04 mg kg^−1^ at Site 1, 0.53 ± 0.03 mg kg^−1^ at Site 2, and 0.46 ± 0.01 mg kg^−1^ at Site 3. The total residues on day 14 were 0.34 ± 0.01 mg kg^−1^ at Site 1, 0.40 ± 0.03 mg kg^−1^ at Site 2, and 0.31 ± 0.02 mg kg^−1^ at Site 3, respectively (Table 4). The half-life ranged from 32 to 41 days, indicating that it remained for a relatively long time when compared to the other test pesticides. Dinotefuran presented a half-life of 1.4–3.1 days and 2.1–2.6 days for Chinese matrimony vine and chili peppers, respectively [19,23], which is much shorter than persimmon, the material used in this study. This can be attributed to the fact that Chinese matrimony vine and red chili peppers present a clear change in the weight due to growth during the pesticide spraying period, but in the case of persimmons, there is minimal change in the weight because the pulp only exhibits ripening rather than growth during the pesticide spraying period. Therefore, the half-life is observed to be longer than that of other crops. Additionally, it has been reported that dinotefuran, which was treated with canola seeds, remained for a long time and exceeded the harvest season [24]. This suggests that even with the same pesticide, the residue period may vary based on the crop and the growing environment. Furthermore, dinotefuran varied based on the site, but metabolites were detected 7 days after the pesticide spraying. This indicates that dinotefuran, a neonicotinoid pesticide, can penetrate into the crop and remain for a long time; it may be metabolized and converted into other substances during that period [24].

The average residual amounts on day 0 of fluazinam, were 0.33 ± 0.01 at Site 1, 0.41 ± 0.03 at Site 2, and 0.36 ± 0.03 mg kg^−1^ at Site 3, and the residual amounts on day 14 were 0.15 ± 0.01 at Site 1, 0.15 ± 0.01 at Site 2, and 0.19 ± 0.01 mg kg^−1^ at Site 3, respectively. The half-lives of fluazinam in the persimmon ranged from 10 to 15 days (Table 4). Wang et al. (2016) reported that the half-life of fluazinam among potatoes was 3.3–5.4 days, and the half-life of cucumbers was 1.0 to 2.5 days [25]. The half-life in mandarin was 8.5–9.5 days [26], and the half-life in grapes was 4.3 days [27]. Fluazinam cannot be easily removed by rain, and thus it is highly likely that the residual half-life is long [28]; this indicates that the residual half-life of fluazinam may vary based on the shape of the crop.

For indoxacarb, the average residual amount was 0.21–0.23 mg kg^−1^ for day 0 and 0.12–0.14 mg kg^−1^ for day 14, and the half-lives were found to be 16 to 20 days (Table 4). The half-life of indoxacarb tends to be relatively long. It may be associated with environmental factors such as evaporation, rainfall, temperature, and sunlight (photodegradation) in the case of open field experiments, and different farming methods and forms may also contribute to changes in the residual amounts [29,30]. It has been reported that the half-life of indoxacarb in the case of cabbages grown in the field is 1.92–2.88 days [31], and the half-life in tomatoes is 3.12 to 3.21 days [32]. Additionally, pomegranate harvested through the process of aging after growth, similar to persimmons, showed a half-life of 7.4–8.4 days, and it is reported that 31–42 days are required to reduce the residual pesticide to the residual acceptance standard of 0.02 mg kg^−1^ [33]. Indoxacarb has a long half-life, unlike other neonicotinoid pesticides, due to its low water solubility (0.2 mg/L) and due to its low rainfall impact [33,34].

The average residual amount of thiacloprid on day 0 and day 14 ranged from 0.18–0.26 mg kg^−1^ and 0.06–0.12 mg kg^−1^, respectively, with a half-life of 9 to 12 days (Table 4). Li et al. (2018) reported that even if the spraying method and throughput were identical, the residual amount and half-life may vary due to the fruit shape, epidermal structure, and decomposition enzyme; the half-life was reported to be 9.55 and 10.6 days, respectively [35]. Additionally, Saimandir and Gopal (2012) reported that the residual amount of pesticides among agricultural products may vary based on whether they exhibit a systemic property; they also reported that thiacloprid can remain in the crop for a relatively long time, and that the half-life of pesticides was 11.1 to 11.6 days [36].

The residual amount of pesticides among agricultural products in the cultivation stage is lost and decomposed by various factors, such as reduction due to crop growth, photolysis, and loss due to rainfall [37]. Several studies have reported that the main factor is the decrease in the residual pesticide amount due to the dilution effect caused by crop growth [38,39]. However, in the case of persimmons, which is a test material, the dilution effect due to growth during pesticide spraying is observed to be minimal. Therefore, it was observed to have a longer half-life than that of other crops. Additionally, it was observed that the residual amount decreased significantly over time (*p* < 0.05). In addition, chromatograms of an analyzed sample with LC-MS/MS were presented in Figure 5.

#### 3.2.2. Residual Concentration of Pesticide on Harvest Day

Table 5 shows the average residual pesticide amount collected on the harvest day according to the Korean PHI.

The dinotefuran metabolite DN was not detected at all the sites, and the combined mean residual amount was 0.35 to 0.46 mg kg^−1^. The residue amounts of dried persimmons, semi-dried persimmons, and dried persimmons prepared using astringent persimmon at Sites 1 and 2 were 0.99–1.03 mg kg^−1^, 0.62–0.65 mg kg^−1^, and 0.89–1.05 mg kg^−1^, respectively.

The residual amounts of fluazinam in fresh persimmons, dried persimmons, and semi-dried persimmons were 0.16–0.18 mg kg^−1^, 0.04–0.06 mg kg^−1^, and 0.03–0.04 mg kg^−1^, respectively, and the dried persimmon slices were below the LOQ.

The residual amounts of indoxacarb were observed to be 0.12–0.14 mg kg^−1^ in fresh persimmon and 0.01–0.04 mg kg^−1^ in the processing products.

The residual amount of thiocloprid, a nonicotinoid pesticide, was 0.06–0.11 mg kg^−1^ for fresh persimmon, and the processing products were in the range of 0.06–0.10 mg kg^−1^, as shown in indoxacarb.

Dinotefuran showed an increasing residual concentration of pesticides during the processing stage, while the other pesticides tended to decrease. The change in the residual amount during the processing stage was determined by calculating the processing and reduction factors.

#### 3.2.3. Processing Factor and Reduction Factor

Residual pesticides are lost or decomposed due to environmental factors and due to the chemical properties of pesticides [37]. Additionally, the concentration of residual pesticides increases or decreases while processing raw materials using various methods, and the absolute amount of pesticides can also decrease. Particularly, food produced by dry processing may have a lower moisture content than raw materials, thereby increasing residual concentration, and the residual concentration in processes such as hot-air drying, may decrease based on the pesticide characteristics [27]. However, pesticides that can be decomposed by heat and highly volatile pesticides are likely to decrease the absolute amount of pesticides if processed by a high-temperature drying method. Therefore, a clearer residual characteristic can be identified in the analysis of the residual pesticide change by processing agricultural products only when the change in the absolute amount of pesticides is analyzed, along with the residual pesticide concentration by processing.

The processing factors of dinotefuran were 1.45 ± 0.27–2.81 ± 0.30, and the residual concentration increased due to processing, as shown in Table 6. The processing factors of thiacloprid lie in the range of 0.71 ± 0.09–0.99 ± 0.25, which is close to 1, and the change in the residual concentration during the processing stage is negligible.

However, the processing factors of both fluazinam and indoxacarb were less than 1, and the concentration of residual pesticides decreased during the process. This suggests that a certain amount of systemic pesticides may remain in the flesh even if the skin of the persimmon is removed and processed.

The reduction factors calculated considering the moisture content that is removed during the processing stage were all dried below 1, and the absolute amount of residual pesticides was decreased. Particularly in the case of dinotefuran, where the processing factor was 1 or more, the reduction factor was found to be less than 1. This result indicates that the residual concentration may increase throughout the processing stage, but the actual absolute amount of pesticides may decrease [11].

## 4. Conclusions

Pesticides, which are sprayed to help in the growth of agricultural products, are decomposed by environmental factors such as rainfall and light, and the amount of pesticides is reduced due to dilution by growth. Additionally, the amount of pesticide is reduced based on the characteristics of the pesticides, and it is metabolized into other substances due to factors such as hydrolysis. Some pesticides can be decomposed even after harvest during the storage process, but the dynamics of the residual pesticides are determined while manufacturing processed foods. The most basic method of processing agricultural products is drying, which is the representative processing method for fruits. In this study, the systemic property of pesticides was determined as an important factor in determining the residual characteristics of processed products by analyzing the behavior of pesticides sprayed onto persimmon during the processing stage after harvest. The absolute amount of pesticides is decreased significantly, while the residual concentration remains high although dinotefuran and thiacloprid, which are systemic pesticides, are used to manufacture processed products after removing the skin of the persimmon. Additionally, the residual amounts of non-systemic pesticides, fluazinam and indoxacarb, were low. These results indicate that the systemic properties of pesticides are an important factor in determining the residual pattern of the processed products. Various processed products are being manufactured to improve both the ease of intake and nutritional function. The main limitation of this study is that it conducts an evaluation to set the MRL and to estimate the risk only for fresh agricultural products. In the future, the residual pesticide movements must be identified for various processed foods to improve the safety of food intake through risk assessment.

## Figures and Tables

**Figure 1 foods-11-00416-f001:**
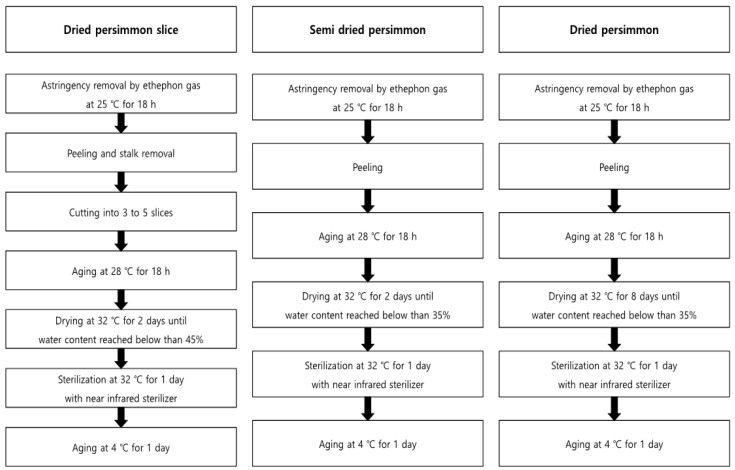
Manufacturing process of persimmon products.

**Figure 2 foods-11-00416-f002:**
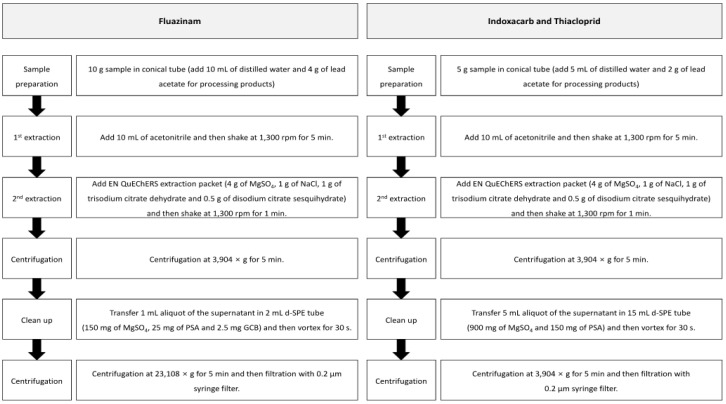
The modified QuEChERS method used for pesticide residue analysis of fluazinam, indoxacarb, and thiacloprid in fresh persimmon and its processing products.

**Figure 3 foods-11-00416-f003:**
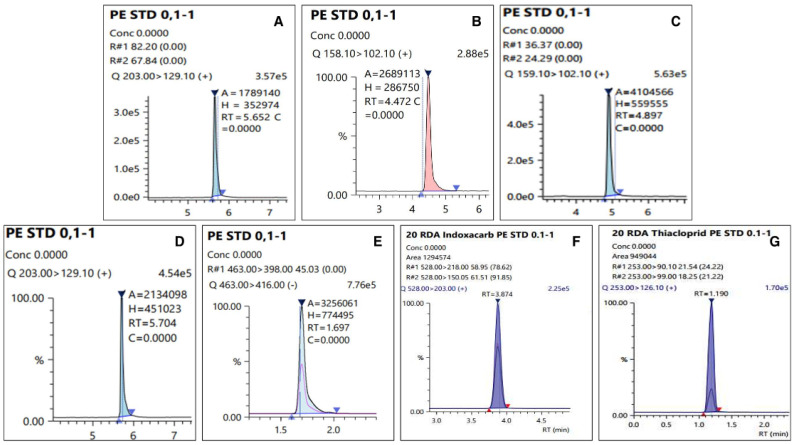
Chromatograms of matrix-matched standard at 100 µg L^−1^: (**A**) Dinotefuran, (**B**) DN, (**C**) MNG, (**D**) UF, (**E**) Fluazinam, (**F**) Indoxacarb, and (**G**) Thiacloprid.

**Figure 4 foods-11-00416-f004:**
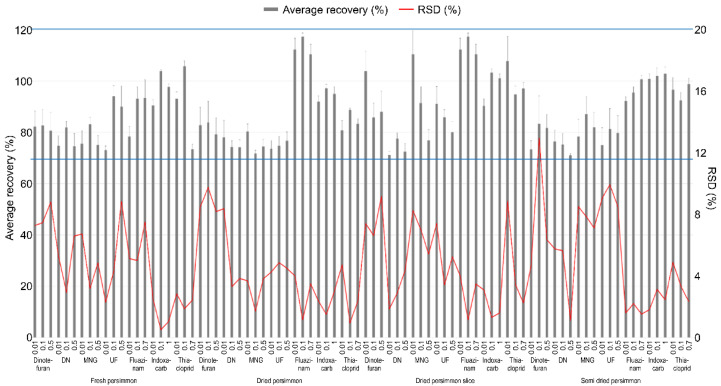
Recovery and RSD of test pesticide in fresh persimmon and its processed products for the validation of the analysis method. The effective range of the recovery is indicated by the blue line.

**Figure 5 foods-11-00416-f005:**
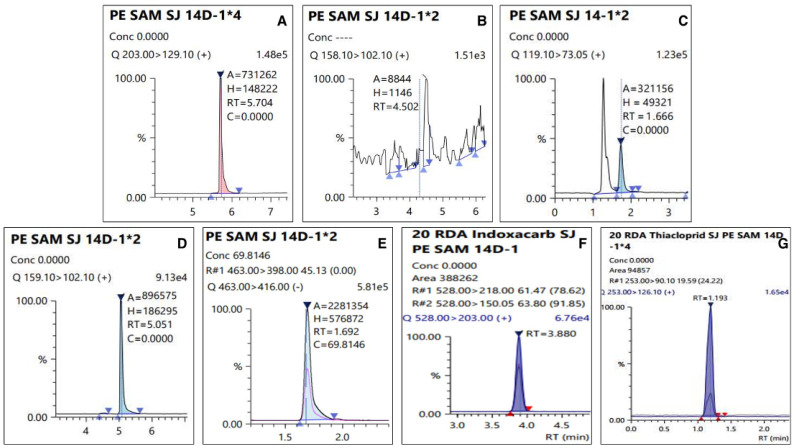
Chromatograms of residual pesticide in the fresh persimmon collected from site 2 on 14 day: (**A**) Dinotefuran (0.24 mg kg^−1^), (**B**) DN (less than LOQ), (**C**) MNG (0.05 mg kg^−1^), (**D**)UF (0.03 mg kg^−1^), (**E**) Fluazinam (0.14 mg kg^−1^), (**F**) Indoxacarb (0.12 mg kg^−1^), and (**G**) Thiacloprid (0.07 mg kg^−1^).

**Table 1 foods-11-00416-t001:** Detailed information of pesticide sprayed onto persimmon.

Pesticide	Site	Application Rate per Treatment											
		Active Ingredient (kg ha^−1^)				Water (L ha^−1^)				Active Ingredient (kg hL^−1^)			
		1st Treatment	2nd Treatment	3rd Treatment	4th Treatment	1st Treatment	2nd Treatment	3rd Treatment	4th Treatment	1st Treatment	2nd Treatment	3rd Treatment	4th Treatment
Dinotefuran	1	0.418	0.469	0.475	-	4185	4689	4751	-	0.00930	0.01042	0.01056	-
	2	0.449	0.536	0.584	-	4489	5360	5841	-	0.00998	0.01191	0.01298	-
	3	0.443	0.439	0.451	-	4433	4386	4513	-	0.00985	0.00975	0.01003	-
Fluazinam	1	0.918	0.933	1.046	1.060	4589	4667	5229	5298	0.02040	0.02074	0.02324	0.02355
	2	1.059	0.882	1.054	1.148	5296	4412	5268	5740	0.02354	0.01961	0.02341	0.02551
	3	0.915	0.927	0.917	0.944	4577	4637	4587	4721	0.02034	0.02061	0.02039	0.02098
Indoxacarb	1	0.259	0.269	0.277	-	5181	5381	5537	-	0.00576	0.00598	0.00615	-
	2	0.278	0.237	0.223	-	5562	4736	4452	-	0.00618	0.00526	0.00495	-
	3	0.227	0.212	0.213	-	4536	4239	4250	-	0.00504	0.00471	0.00472	-
Thiacloprid	1	0.251	0.237	0.236	-	5018	4743	4723	-	0.00558	0.00527	0.00525	-
	2	0.266	0.232	0.238	-	5327	4640	4766	-	0.00592	0.00516	0.00530	-
	3	0.222	0.179	0.193	-	4433	3583	3863	-	0.00493	0.00398	0.00429	-

**Table 2 foods-11-00416-t002:** Multiple reaction monitoring (MRM) conditions of each target pesticide for LC-MS/MS analysis.

Compound	Precursor (*m*/*z*)	Quantitation		Confirmation	
		*m/z*	CE (eV)	*m/z*	CE (eV)
Dinotefuran	203.00	129.10	13	114.10	13
DN	158.10	102.10	16	57.05	24
MNG	119.10	73.05	12	44.00	23
UF	159.10	102.10	13	67.00	20
Fluazinam	463.00	416.00	19	398.00	18
Indoxacarb	528.00	203.00	−40	150.00	−23
Thiacloprid	253.00	126.10	−21	90.1	−37

**Table 3 foods-11-00416-t003:** Correlation coefficient (r^2^) of test pesticides according to matrices for quantitation in the samples.

Pesticide	Study	Matrix	Linear Range (µg L^−1^)	r^2^
Dinotefuran	Harvest	Fresh persimmon	1–100	0.9965
		Dried persimmon		0.9980
		Dried persimmon slice		0.9989
		Semi dried persimmon		0.9967
	Decline	Fresh persimmon		0.9995
DN (Dinotefuran metabolite)	Harvest	Fresh persimmon	1–100	0.9994
		Dried persimmon		0.9987
		Dried persimmon slice		0.9998
		Semi dried persimmon		0.9998
	Decline	Fresh persimmon		0.9998
MNG (Dinotefuran metabolite)	Harvest	Fresh persimmon	1–100	0.9984
		Dried persimmon		0.9993
		Dried persimmon slice		0.9987
		Semi dried persimmon		0.9986
	Decline	Fresh persimmon		0.9987
UF (Dinotefuran metabolite)	Harvest	Fresh persimmon	1–100	0.9972
		Dried persimmon		0.9998
		Dried persimmon slice		0.9995
		Semi dried persimmon		0.9981
	Decline	Fresh persimmon		0.9992
Fluazinam	Harvest	Fresh persimmon	2–100	0.9992
		Dried persimmon		0.9996
		Dried persimmon slice		0.9997
		Semi dried persimmon		0.9996
	Decline	Fresh persimmon		0.9993
Indoxacarb	Harvest	Fresh persimmon	1–100	1.0000
		Dried persimmon		1.0000
		Dried persimmon slice		0.9999
		Semi dried persimmon		1.0000
	Decline	Fresh persimmon		0.9999
Thiacloprid	Harvest	Fresh persimmon	1–100	0.9996
		Dried persimmon		0.9999
		Dried persimmon slice		0.9997
		Semi dried persimmon		0.9997
	Decline	Fresh persimmon		0.9999

**Table 4 foods-11-00416-t004:** Dissipation residue of test pesticide and their half-lives in fresh persimmon in site 3.

Site	DALA ^1^	Mean Residue ± SD ^2^) (*n* = 3, mg kg^−1^)								Half-Life (day)			
		Dinotefuran					Fluazinam	Indoxacarb	Thiacloprid	Dinotefuran	Fluazinam	Indoxacarb	Thiacloprid
		Parent	DN	MNG	UF	Total							
1	0	0.47 ± 0.04	<0.01 ^3^	<0.01	<0.01	0.47 ± 0.04 ^a 4^	0.33 ± 0.01 ^a^	0.23 ± 0.01 ^a^	0.26 ± 0.02 ^a^	36	11	20	12
	1	0.41 ± 0.03	<0.01	<0.01	<0.01	0.41 ± 0.03 ^b^	0.32 ± 0.02 ^a^	0.22 ± 0.02 ^ab^	0.26 ± 0.01 ^a^				
	3	0.38 ± 0.02	<0.01	<0.01	<0.01	0.38 ± 0.02 ^bc^	0.28 ± 0.03 ^b^	0.20 ± 0.01 ^bc^	0.25 ± 0.04 ^a^				
	5	0.37 ± 0.02	<0.01	<0.01	<0.01	0.37 ± 0.02 ^bc^	0.21 ± 0.01 ^c^	0.18 ± 0.01 ^cd^	0.20 ± 0.01 ^b^				
	7	0.34 ± 0.02	<0.01	<0.01	0.01 ± 0.00	0.35 ± 0.02 ^c^	0.19 ± 0.01 ^d^	0.17 ± 0.02 ^d^	0.16 ± 0.03 ^c^				
	14	0.28 ± 0.01	<0.01	0.02 ± 0.01	0.03 ± 0.01	0.34 ± 0.01 ^c^	0.15 ± 0.01 ^e^	0.14 ± 0.01 ^e^	0.12 ± 0.01 ^c^				
2	0	0.53 ± 0.03	<0.01	<0.01	<0.01	0.53 ± 0.03 ^a^	0.41 ± 0.03 ^a^	0.23 ± 0.01 ^a^	0.22 ± 0.05 ^a^	41	10	16	10
	1	0.47 ± 0.04	<0.01	<0.01	<0.01	0.47 ± 0.04 ^ab^	0.40 ± 0.01 ^a^	0.21 ± 0.01 ^b^	0.22 ± 0.04 ^a^				
	3	0.45 ± 0.04	<0.01	<0.01	<0.01	0.45 ± 0.04 ^b^	0.34 ± 0.00 ^b^	0.18 ± 0.02 ^c^	0.20 ± 0.03 ^ab^				
	5	0.42 ± 0.04	<0.01	<0.01	<0.01	0.42 ± 0.04 ^b^	0.31 ± 0.01 ^c^	0.17 ± 0.01 ^c^	0.17 ± 0.04 ^ab^				
	7	0.38 ± 0.03	<0.01	0.02 ± 0.01	0.01 ± 0.01	0.41 ± 0.04 ^b^	0.23 ± 0.02 ^d^	0.15 ± 0.01 ^d^	0.15 ± 0.02 ^b^				
	14	0.24 ± 0.02	<0.01	0.07 ± 0.02	0.03 ± 0.01	0.40 ± 0.03 ^b^	0.15 ± 0.01 ^e^	0.12 ± 0.01 ^e^	0.08 ± 0.02 ^c^				
3	0	0.46 ± 0.01	<0.01	<0.01	<0.01	0.46 ± 0.01 ^a^	0.36 ± 0.03 ^a^	0.21 ± 0.01 ^a^	0.18 ± 0.01 ^a^	32	15	17	9
	1	0.37 ± 0.04	<0.01	<0.01	<0.01	0.37 ± 0.04 ^b^	0.35 ± 0.01 ^a^	0.21 ± 0.01 ^a^	0.17 ± 0.01 ^ab^				
	3	0.35 ± 0.03	<0.01	<0.01	<0.01	0.35 ± 0.03 ^bc^	0.35 ± 0.01 ^a^	0.19 ± 0.01 ^b^	0.16 ± 0.02 ^b^				
	5	0.33 ± 0.02	<0.01	<0.01	<0.01	0.33 ± 0.02 ^bc^	0.34 ± 0.01 ^a^	0.16 ± 0.01 ^c^	0.11 ± 0.02 ^c^				
	7	0.29 ± 0.03	<0.01	<0.01	0.02 ± 0.00	0.32 ± 0.03 ^c^	0.28 ± 0.02 ^b^	0.15 ± 0.01 ^c^	0.09 ± 0.01 ^c^				
	14	0.27 ± 0.03	<0.01	<0.01	0.04 ± 0.01	0.31 ± 0.02 ^c^	0.19 ± 0.01 ^c^	0.12 ± 0.01 ^d^	0.06 ± 0.01 ^e^				

^1^ Day after last application; ^2^ Standard deviation; ^3^ Less than limit of quantitation; ^4^ Different letters in the column indicate significant differences at *p* < 0.05 by least significant deviation.

**Table 5 foods-11-00416-t005:** Average residue of test pesticides in fresh persimmon and their processed products on harvest day.

Matrix	Site	Mean Residue ± SD ^1^ (*n* = 3, mg kg^−1^)							
		Dinotefuran					Fluazinam	Indoxacarb	Thiacloprid
		Parent	DN	MNG	UF	Total			
Fresh persimmon	1	0.30 ± 0.04	<0.01 ^2^	0.02 ± 0.02	0.02 ± 0.01	0.35 ± 0.02	0.16 ± 0.02	0.14 ± 0.01	0.11 ± 0.01
	2	0.25 ± 0.03	<0.01	0.10 ± 0.02	0.03 ± 0.01	0.46 ± 0.05	0.16 ± 0.02	0.12 ± 0.01	0.08 ± 0.02
	3	0.28 ± 0.03	<0.01	<0.01	0.04 ± 0.01	0.33 ± 0.02	0.18 ± 0.02	0.12 ± 0.01	0.06 ± 0.01
Dried persimmon	1	0.77 ± 0.07	<0.01	0.05 ± 0.01	0.10 ± 0.01	0.99 ± 0.09	0.06 ± 0.00	0.04 ± 0.01	0.10 ± 0.01
	2	0.72 ± 0.04	<0.01	0.11 ± 0.01	0.10 ± 0.01	1.03 ± 0.04	0.04 ± 0.00	0.03 ± 0.01	0.08 ± 0.01
Semi-dried persimmon	1	0.50 ± 0.03	<0.01	0.02 ± 0.01	0.07 ± 0.01	0.62 ± 0.04	0.04 ± 0.01	0.03 ± 0.01	0.08 ± 0.02
	2	0.46 ± 0.04	<0.01	0.07 ± 0.01	0.06 ± 0.00	0.65 ± 0.06	0.03 ± 0.01	0.02 ± 0.00	0.07 ± 0.01
Dried persimmon slice	1	0.65 ± 0.03	<0.01	0.07 ± 0.01	0.09 ± 0.00	0.89 ± 0.05	<0.01	0.02 ± 0.01	0.09 ± 0.01
	2	0.66 ± 0.02	<0.01	0.15 ± 0.02	0.1 ± 0.03	1.05 ± 0.05	<0.01	0.01 ± 0.00	0.06 ± 0.02

^1^ Standard deviation, ^2^ Less than limit of quantitation.

**Table 6 foods-11-00416-t006:** Processing factor and reduction factor of test pesticides caused by processing.

Site	Matrix	Dinotefuran		Fluazinam		Indoxacarb		Thiacloprid	
		PF ^1^	RF ^2^	PF	RF	PF	RF	PF	RF
1	Dried persimmon	2.81 ± 0.30	0.80 ± 0.09	0.39 ± 0.04	0.11 ± 0.01	0.32 ± 0.07	0.08 ± 0.02	0.91 ± 0.08	0.24 ± 0.02
	Dried persimmon slice	1.76 ± 0.17	0.83 ± 0.07	-	-	0.12 ± 0.04	0.04 ± 0.01	0.79 ± 0.04	0.24 ± 0.01
	Semi-dried persimmon	2.53 ± 0.23	0.80 ± 0.08	0.23 ± 0.07	0.11 ± 0.03	0.22 ± 0.08	0.10 ± 0.04	0.76 ± 0.17	0.36 ± 0.08
2	Dried persimmon	2.28 ± 0.29	0.64 ± 0.08	0.25 ± 0.02	0.07 ± 0.01	0.27 ± 0.03	0.08 ± 0.01	0.99 ± 0.25	0.29 ± 0.07
	Dried persimmon slice	2.31 ± 0.26	0.75 ± 0.08	-	-	0.08 ± 0.00	0.03 ± 0.00	0.71 ± 0.09	0.25 ± 0.03
	Semi-dried persimmon	1.45 ± 0.27	0.64 ± 0.12	0.16 ± 0.04	0.07 ± 0.02	0.16 ± 0.01	0.09 ± 0.00	0.89 ± 0.10	0.24 ± 0.05

^1^ Processing factor, ^2^ Reduction factor.

## Data Availability

Data is contained within the article.

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
