# Peer review of "Dissipation and Residue Pattern of Dinotefuran, Fluazinam, Indoxacarb, and Thiacloprid in Fresh and Processed Persimmon Using LC-MS/MS"

_foods, 2022, doi:10.3390/foods11030416_

Round 1
Reviewer 1 Report
The article deals with the behavior of 4 pesticides after they are applied to persimmons. The study is interesting and a lot of work has been done on it.
The article is a very interesting topic from the point of view of pesticides, unfortunately some parts are written in very poor English and it is difficult to follow the author's idea. There are also many stylistic errors in the article. What I miss in the article is whether the samples came from one tree or from several trees. How large was the sample that was homogenized, whether it was a single fruit or multiple fruits. Nowhere in the article did I find whether they dealt with the effects of the weather (whether the sun was shining, it was raining, the wind was blowing), which is important in such a study.
Here are some comments:
line 37: Pesticides are a group of chemical substances which are further divided into fungicides, insecticides, herbicides, etc.
lines 46 - 47 This sentence does not make sense. I think that correct is: EU-harmonized MRLs are set for more than 1,300 pesticides covering 378 food products / food groups. A default MRL of 0.01 mg / kg is applicable to nearly 690 of these pesticides which are not explicitly mentioned in the MRL legislation. (https://doi.org/10.2903/j.efsa.2021.6491)
line 56-57: wrong word order
line 70: after ??? various drying processes
line 100: tested pesticides ??
line 102: what does it mean that pesticides have been sprayed 2000 times?
line 111-113: I don't understand this sentence.
paragraph 2.2 reagents and materials: rotary evaporator is missing
line 142: I assume that the weight of both fresh and processed (dried) sample was 5 g and was each separate?
line 148: What water was used? distilled, deionized
line 151 - 153: duplicate sentences. In both, only the information that QuEChERS was used is listed. I recommend rewriting section 2.4 completely
line 165 pesticide residues were analyzed ??? line 167 - 168: The column for the analysis of dinotefuran .. in the samples was ....
line 170 - 171 described column was used for the analysis of these pesticides, not the other way around. line 177: really stabilization of initial conditions took 100 min ???
line 188: was the source temperature really only 150 ° C?
line 191: Table 2 shows the monitored MRM transitions for each analyte. Table 2: in the description of the paragraph the quantification ion is an error (and since MS / MS is used so the transitions and not the ion are monitored)
paragraph 2.5 method validation: Here the author states 2016 as a document for comparison of validation parameters, but in the text he has SANTE / 12682/2019. From my point of view, it is definitely more appropriate to mention the SANTE document, a new one from 2021 (must also be updated on line 257 and in the references).
Paragraph 3.1: incorrect rounding (rounded to 3 valid digits): 73.0 - 105.8 ... correct to 73.0 - 106; 0.51 - 8.85 fix to 0.510-8.85 etc.
paragraph 3.2: I would recommend to simplify and clarify this whole paragraph. The information is repeated in the paragraph (eg lines 276-277 and 286-287), most of which are listed in a table.
Table 4 is too large and therefore very confusing. It does not explain what the letters a-e mean. Somewhere the uncertainty value is 0.0. Perhaps in this case it would be better to graphically represent the decrease in the amount of individual analytes monitored and to put these data into a supplementary. There is an interesting big difference in half-life between the individual orchards. What is the difference?
Table 5: Somewhere the uncertainty value is 0.00. In footnote correct the world limit.
Table 6: The same as table 5: uncertainty value is 0.00.
paragraph 3.2.2. : Simplify the paragraph again, correct all pesticide names (line 377)
line 375: dried persimmon slices were not below the LOQ, but the residues in this material were below LOQ. It may be appropriate to mention at the outset that these are two systemic and two contact pesticides, as their behavior is greatly affected. While the contacts remain on the surface of the fruit, the systemic ones enter the flesh and then it depends on their distribution whether they are more concentrated in the flesh or in the skin. Again correct the name of the pesticide: line 401.
Conclusions: Provide precise summary conclusions of the study with an explanation. The conclusion written in this way is completely vague.
Reviewer 2 Report
A study investigating pesticide residues in Korean agricultural products was presented. In this study, the authors combined QuEchERS and LC/MS/MS analysis of pesticide residues in fresh and processed persimmons. The study found that concentrations of non-systemic pesticides, fluazinam and indoxacarb, decreased in the samples. For systemic pesticides, dinotefuran, residual concentration is increased, and the absolute amount of pesticides is decreased through processing. This research method validates the half-life of systemic pesticides commonly used in agriculture or other applications. The results help the correct use of pesticides and protect ordinary people from excessive intake of pesticides.
The minor editing errors in table 4, 5 and 6 need to be corrected before publishing.
Author Response
I recognized that I had to revise the table and revised the manuscript.
Reviewer 3 Report
Detailed comments are included in the review file attached below.
